# How Czech Adolescents Perceive Their Physical Activity

**DOI:** 10.3390/children10071134

**Published:** 2023-06-29

**Authors:** Ferdinand Salonna, Michal Vorlíček, Lukáš Rubín, Jana Vašíčková, Josef Mitáš

**Affiliations:** 1Institute of Active Lifestyle, Faculty of Physical Culture, Palacký University, 771 11 Olomouc, Czech Republic; michal.vorlicek@upol.cz (M.V.); lukas.rubin@upol.cz (L.R.); josef.mitas@upol.cz (J.M.); 2Department of Physical Education and Sport, Faculty of Science, Humanities and Education, Technical University of Liberec, 461 17 Liberec, Czech Republic; 3Department of Social Sciences in Kinanthropology, Faculty of Physical Culture, Palacký University, 771 11 Olomouc, Czech Republic; jana.vasickova@upol.cz

**Keywords:** MVPA, VPA, social norms, adolescents, perception

## Abstract

Adolescence is a critical stage in the development of an individual’s physical activity (PA) habits and preferences. Adolescents’ perceptions of PA can influence their motivation to engage in PA and, consequently, their overall level of PA. Thus, our primary aim was to investigate whether Czech adolescents misperceive their peers’ PA. Our dataset comprised cross-sectional data on 1289 adolescents aged 11–15 years. PA was measured using self-reported items used in the HBSC study. To describe the gender and school grade differences in VPA, independent samples T, ANOVA, Mann–Whitney U, and Kruskal–Wallis H tests were performed. To analyze the effect of gender, school grade, school, and participants’ own PA on the underestimation of PA, binomial regression models were used. Our study points out that there is a discrepancy between self-perceived levels of PA and the perceived descriptive norms of peers’ PA. Adolescents underestimate the prevalence of sufficient PA, and thus perceived descriptive norms in PA as being worse than levels of own PA. These findings indicate room for targeted interventions based on social-norms-based approaches to increase the PA of adolescents or at least strengthen their actual positive behavior.

## 1. Introduction

Engaging in sufficient physical activity (PA) can have significant benefits for the overall health of children and adolescents. This includes improvements in their physical well-being, as well as positive impacts on their mental and social–emotional health [1]. Despite this, studies indicate that adolescents are failing to meet the recommended levels of PA [2]. As with adolescents in other countries, Czech adolescents are facing the same issue, as a significant proportion of them do not meet public health guidelines for recommended levels of PA [3].

The conduct of individuals regarding their health is significantly impacted by social norms. Social norms refer to the shared standards, expectations, and rules of behavior that are considered acceptable and appropriate within a particular social group or society. These norms guide and regulate individual as well as group behavior, defining what is considered normal, acceptable, and desirable in terms of actions, attitudes, values, and beliefs. Thus, perceiving what is “normal” influences individuals’ behavior. Such influence ranges from reinforcing favorable behaviors that can safeguard and improve one’s well-being to promoting unfavorable actions that elevate the possibility of detrimental health outcomes [4].

The development of a theoretical basis for the investigation of social norms is grounded in the evolution of several frameworks. Initially, social norms were defined and explored within larger theories, such as the theory of reasoned action and the theory of planned behavior [5]. The focus theory of normative conduct evolved from the theory of planned behavior [6], focusing exclusively on social norms.

The theory of normative conduct framework delineates norms into two distinct categories: injunctive and descriptive [7]. Injunctive norms involve an evaluative component, whereas descriptive norms do not. Descriptive norms refer to individuals’ perceptions of common behaviors within their social context. When faced with unfamiliar situations, people may rely on descriptive norms as a shortcut for understanding socially appropriate behavior. By observing the actions of others around them, and if widespread participation indicates social acceptability, individuals may adopt similar conduct themselves.

As the study of social norms advanced, researchers began to consider referent groups for injunctive and descriptive norms in terms of proximity: either proximal or distal. Proximal refers to an individual’s close friend circle (for example, a few friends that they spend most of their time with). On the other hand, distal pertains to a larger population, such as students at one’s university [8,9].

As the research on social norms has progressed, correlational studies have provided compelling evidence that descriptive norms have a positive association with both intention and behavior. The effectiveness of norms as a means of influencing behavior has been demonstrated across a wide range of behaviors, such as promoting healthier eating habits, encouraging energy conservation, and reducing binge drinking [10,11]. Studies investigating the connection between social norms and PA are limited [12]. Even though research on social norms has not placed much emphasis on PA, certain studies demonstrate a correlation between the two [4,13,14,15,16,17]. Previous difficulty in linking subjective norms to PA intentions might be due to deficient methods of measuring and analyzing subjective norms [12].

Gender differences in PA among adolescent boys and girls are multifaceted and intertwined with societal norms, gender roles, and self-presentation [18]. During the transition from childhood to adolescence, peer influence increases and plays a crucial role in engagement in PA [19]. This influence can comprise social support, the presence of peers during PA, peer norms, friendship quality, changes to friendship groups, preferences for certain activities among peers, and affiliation with peer groups [20]. Additionally, potential experiences of peer victimization may further limit girls’ participation in PA [21]. Despite being a central factor affecting girls’ PA involvement, the influence of peers has often been overlooked by many interventions aimed at promoting PA [18].

Previous studies have shown that people tend to overestimate the prevalence of negative behaviors and attitudes among their peers while underestimating positive ones. This misperception can lead to negative behaviors, such as excessive alcohol consumption, and a decrease in positive behaviors, such as healthy eating, using sun protection, and being physically active [11,22]. The social norms approach is commonly used to encourage positive health behaviors by addressing individuals’ misperceptions of their peers’ attitudes and actions [23]. In addition, the findings of Vorlíček et al. [24] on a sample of Czech adolescents indicate that there might be room for targeted interventions based on the social norms approach to increase the PA of adolescents or at least strengthen their actual positive behavior.

Gaining insights into how adolescents perceive their PA is essential for the development of effective interventions and strategies aimed at promoting a more active lifestyle. Understanding the disparity between individuals’ self-perceived levels of PA and their perceptions of social norms is particularly important when applying the social norms approach. Previous research has devoted relatively limited attention to this specific area, underscoring the significance of our study as a valuable addition to the existing literature. By addressing this research gap, our study provides novel insights and contributes to a more comprehensive understanding of the subject matter. Therefore, our main objective was to examine whether Czech adolescents have misconceptions regarding their peers’ physical activity behaviors. Additionally, we aimed to explore the relationships between their self-perceived physical activity, descriptive social norms of physical activity, and whether these associations varied based on gender and grade level.

## 2. Materials and Methods

### 2.1. Procedure Sample and Participant Selection

Our cross-sectional study involved 1289 adolescents from the Czech Republic. The participants were aged between 11 and 15 years, with a mean age of 13.49 years and a standard deviation of 1.16 years. Out of the total participants, 46% were boys. The study included 12 randomly selected schools, which comprised students from grades 6 to 9. Schools specializing in sports and schools for students with special educational needs were not included in the study. Within each school we randomly selected one class per grade. It was assumed that the classes within each school were approximately equal in size. This sampling approach aimed to create a self-weighted sample, wherein each student had an approximately equal chance of being selected for participation.

Participation in the study was voluntary, meaning that the adolescents could choose whether or not to participate. No incentives were provided to encourage participation, and the data collection process did not involve any form of compensation or reward. Additionally, the study reported that less than five percent of the adolescents opted out of participating, indicating a relatively high rate of participation among the targeted population. 

### 2.2. Assessments and Measures

For the purposes of this study, we used questions on MVPA and VPA from the HBSC study [25]. According to a current systematic review [26], PA measures of the HBSC questionnaire are moderately reliable (ICC 0.41–0.6). When assessing validity, the HBSC PA measures show fair to moderate performance (r 0.21–0.6).

### 2.3. Procedure

The data were collected from 2017 to 2018 in regular school weeks during the spring and autumn seasons. The pupils filled in an electronic questionnaire at school during class under the supervision of teachers and researchers.

### 2.4. Data Processing

To describe the gender and school grade differences, an independent samples T test and ANOVA were performed for MVPA; a Mann–Whitney U test and Kruskal–Wallis H test were performed for VPA. To analyze the effect of gender, school grade, school, and participants’ own PA on the underestimation of PA, binomial regression models were used. The sample was dichotomized into two groups: underestimated PA (1) vs. rest of the sample (0). In four steps we entered sex, grade, school, and own PA. Statistical data processing was performed using IBM SPSS Statistics software, version 23 (IBM, Armonk, NY, USA).

## 3. Results

### 3.1. MVPA

According to results showed in Table 1, 18% of Czech adolescents reported more than 60 min of MVPA daily. Significantly more boys than girls (20.6% vs. 15.8%) had 60 min of daily MVPA (χ^2^ = 22.21; *p* = 0.002). Levels of MVPA gradually and significantly decreased from 6th grade to 9th grade (χ^2^ = 50.927; *p* < 0.001). While in 6th grade 23.9% of adolescents had more than 60 min of daily MVPA, in 7th grade it was 21.9%, in 8th grade it was 16.8%, and in 9th grade it was only 8.4%.

Regarding perceived descriptive norms in MVPA, only 3.9% of adolescents believed that their peers have at least 60 min of daily MVPA (Table 2). Boys perceived their peers as having more MVPA compared to girls (χ^2^ = 27.435; *p* = 0.001). We did not observe significant differences across grades (χ^2^ = 31.954; *p* = 0.059).

When comparing self-perceived MVPA and perceived descriptive norms in MVPA (i.e., adolescents’ own behavior vs. the behavior of their peers), we found clear misperception (Figure 1). Adolescents perceive the level of MVPA of their peers as much lower. While the majority of adolescents (57.7%) have reported 60 min of MVPA at least 4 times a week, only 35.9% believe that their peers have 60 min of MVPA at least 4 times a week. The pattern of misperception is present both for boys and girls and across all grades (Table 2 and Table 3).

In order to identify which adolescents misperceive MVPA, we performed a binominal logistic regression. The sample was dichotomized into two groups: underestimated MVPA (1) vs. the rest of the sample (0). In four steps we entered sex, grade, school, and own MVPA.

According to the results, girls and older students tend to underestimate the prevalence of MVPA significantly more among their peers. School had no effect on the perception of MVPA. After entering the self-perceived MVPA into the model, almost all of the variance is explained and most of the gradient disappeared (Table 4). Adolescents who have 0 days of MVPA weekly underestimate the MVPA of their peers significantly more compared to their peers who have 60 min daily of MVPA (OR 1.525). The more passive adolescents are, the more they tend to perceive passive behavior as the norm and vice versa, the more active adolescents tend to perceive more active behavior as the norm.

### 3.2. VPA

According to the results shown in Table 5, 29.8% of Czech adolescents reported VPA at least 4 times a week. Significantly more boys than girls had VPA at least 4 times a week (U = 186,135; *p* = 0.003). Levels of VPA gradually and significantly decreased from 6th grade to 9th grade (H = 27.929; *p* < 0.001). While in 6th grade 13% of adolescents had daily VPA, in 7th grade it was 10%, in 8th grade it was 8%, and in 9th grade it was only 4%.

Regarding perceived descriptive norms in VPA, only 10% of adolescents believed that their peers engaged in VPA everyday (Table 6). Boys perceived their peers as having more VPA compared to girls (U = 192,695; *p* = 0.048). Levels of VPA gradually and significantly decreased from 6th grade to 9th grade (H = 22.478; *p* < 0.001).

When comparing self-perceived VPA and perceived descriptive norms in VPA (i.e., adolescents’ own behavior vs. the behavior of their peers) we found clear misperception. (Figure 2). While almost 30% of adolescents have reported VPA at least four times a week, only 10% believe that their peers have engaged in VPA at least four times a week. This misperception is present for both boys and girls and across all grades (Table 5 and Table 6).

In order to identify which adolescents misperceive VPA, we performed a binominal logistic regression. The sample was dichotomized into two groups: those that underestimated VPA (1) vs. the rest of the sample (0). In four steps we entered sex, grade, school, and own VPA.

In general, the results were very similar to what we reported for MVPA above, although less expressive. Girls and older students tend to underestimate the prevalence of VPA among their peers to a significantly greater degree. School had no effect on perceptions of VPA. After entering the self-perceived VPA into the model, almost all of the variance is explained and most of the gradient disappeared (Table 7). Adolescents who had no VPA during the last month underestimated the VPA of their peers more than nine times compared to their peers who had daily VPA (OR 3.134). The more VPA adolescents have, the more they tend to perceive more VPA as the norm and vice versa, the less active adolescents tend to perceive less active behavior as the norm.

## 4. Discussion

Our primary aim was to investigate whether Czech adolescents misperceive their peers’ PA behaviors. Furthermore, we aimed to investigate the associations between their self-perceived PA and descriptive social norms PA, and whether these associations differed with gender and class grade. Thus, we were able to evaluate the suitability of utilizing the social norms approach as an intervention strategy for promoting sufficient PA among adolescents.

Our research highlights a disparity between individuals’ perceptions of their own PA levels and their perceptions of their peers’ PA levels. Adolescents tend to underestimate the prevalence of adequate PA among their peers, resulting in perceived descriptive norms that indicate lower levels of PA compared to their own. Girls and older students demonstrate a notable tendency to underestimate the prevalence of PA among their peers. Interestingly, the school environment does not appear to have an impact on perceptions of PA. When individuals’ own self-perceived levels of PA are taken into account, it becomes evident that they explain a significant portion of the variability in misperceptions related to gender and school grade. Self-perceived PA tends to account for nearly all of the differences observed among genders and school grades in terms of misperceptions.

Our study also underscores a notable disparity between individuals’ self-perceived levels of MVPA and VPA and their perceptions of their peers’ physical activity norms. Despite the majority of adolescents (57.7%) reporting their engagement in at least four 60 min sessions of MVPA per week, a significantly lower percentage (35.9%) perceive their peers as achieving the same activity level. Importantly, this pattern of misperceptions extends to VPA as well, affecting both boys and girls across all grades. These findings have significant implications for the utilization of social norms in intervention strategies aimed at promoting sufficient PA among adolescents. Perkins [27] has identified two pre-conditions that must be met before implementing the social norms approach properly. The first condition necessitates a discrepancy between perceived and actual behaviors, which indicates a problem of the overestimation of conduct. The second condition demands that at least half of the population behaves responsibly; given that individuals strive to conform to societal norms, employing a social norms message campaign may inadvertently encourage harmful conduct if most people engage in it. Hence, if more than 50% violate desirable behaviors targeted by the intervention program, recognizing the limitations of the social norms approach would prove critical in decision making regarding its implementation potential [11,22,28].

In line with numerous studies before [29,30], our findings confirmed that girls report lower levels of PA. In addition, according to our findings girls tend to underestimate the prevalence of VPA among their peers to a significantly greater extent. In their comprehensive review of existing systematic reviews, Duffey et al. [31] examine the barriers and facilitators of PA specifically in adolescent girls. Their findings indicate that the most commonly reported barriers to PA among adolescent girls were the absence of support from peers, family, and teachers, closely followed by time constraints. On the other hand, the most frequently identified facilitators of PA were weight loss/management goals and the support provided by peers, family, and teachers. It is worth noting that the majority of factors influencing PA engagement among girls primarily originated from the individual and interpersonal levels. Our results confirm their findings and indicate that social norms are an important component in facilitating PA among adolescent girls.

We confirmed a clear relationship between descriptive norms and PA. According to our findings, the more passive that adolescents are, the more they tend to perceive the norm of this behavior as being less frequent. In previous literature, there is general support for the notion that social norms can influence individual decisions to be physically active [4]. Several studies have explored the correlation between descriptive social norms and PA, concluding that the perceived behavior of one’s peers strongly influenced their likelihood to engage in PA [32,33]. Ball et al. [32] also concluded that descriptive norms had an impact separate from social support and proposed adjusting social norms in interventions aimed at encouraging higher levels of PA.

We found that grade had an effect on misperceptions of MVPA and VPA. On the contrary, school had no effect. If we may define a school grade as a more proximal and school as a more distal norm, our findings would be in line with previous research, which indicates that norms closer to an individual have a greater impact on their behavior than those further away. For example, studies by Yun and Silk [9] showed that proximal descriptive and injunctive norms had more predictive power in promoting healthy food choices compared to distal norms. Likewise, Korcuska and Thombs [34] found that college students’ alcohol use was more influenced by the perceived normative behaviors of close friends than those of the typical student.

In general, we found that, unfortunately, a very low number of adolescents fulfilled the recommendations for sufficient physical activity, both MVPA and VPA. We also confirmed that boys are more active than girls and that PA levels gradually decrease with older age. These findings are fully in line with other Czech [3,35,36] studies, but also with wider regional [37,38] and global situations [39]. Despite the significant health benefits associated with regular PA, research has indicated that numerous adolescents fail to meet PA recommendations [29,30]. Such non-compliance could be attributed to several underlying factors, such as technological advancements, changes in modern diets, and reduced outdoor play, among others [31].

The study has a number of strengths and limitations that must be considered. A limitation worth noting is that the results rely on self-reported data, which may produce reporting biases or various alternative explanations (such as social pressure to respond in particular ways). Our participants reported their own PA levels along with those of their peers. Additionally, due to our cross-sectional approach, we cannot conclude any causal relationship between actual PA levels and perceived norms among adolescents. Another limitation could be that we focused on solely descriptive norms. It was revealed that modifying descriptive norms alone may not be sufficient for changing behavior [40]. The direct impact of descriptive norms on actions is considered unlikely, as there are likely moderators or mediators involved between these norms and conduct. In this context, injunctive norms, outcome expectations, and group identity have been identified as key normative mechanisms [41], while peer communication and issue familiarity have also received some attention in related research studies [11,42].

Nevertheless, this study emphasizes the importance of examining adolescents’ physical activities within the context of their perceptions of social norms related to physical activity. This specific area has received relatively less attention in previous research, making this study a valuable contribution to the existing literature. To the best of our knowledge, no study investigating this specific topic has been conducted in the Czech Republic or Central Europe. Thus, our study fills an important research gap by being the first of its kind in this region, providing valuable insights into the perceptions of adolescents’ physical activities and their relationship with social conventions. By exploring this unexplored territory, our research expands the existing knowledge base and contributes to a more comprehensive understanding of the factors influencing adolescents’ physical activity behaviors in the Czech Republic and Central Europe.

## 5. Conclusions

Our study points out that there is a discrepancy between the self-perceived levels of own MVPA and VPA and the perceived descriptive norms of peers’ MVPA and VPA. Adolescents underestimate the prevalence of sufficient MVPA and VPA, and thus perceived descriptive norms in MVPA and VPA are worse than levels of own MVPA and VPA. These findings indicate room for targeted interventions based on the social-norms-based approaches to increase the physical activity of adolescents or at least strengthen their actual positive behaviors.

## Figures and Tables

**Figure 1 children-10-01134-f001:**
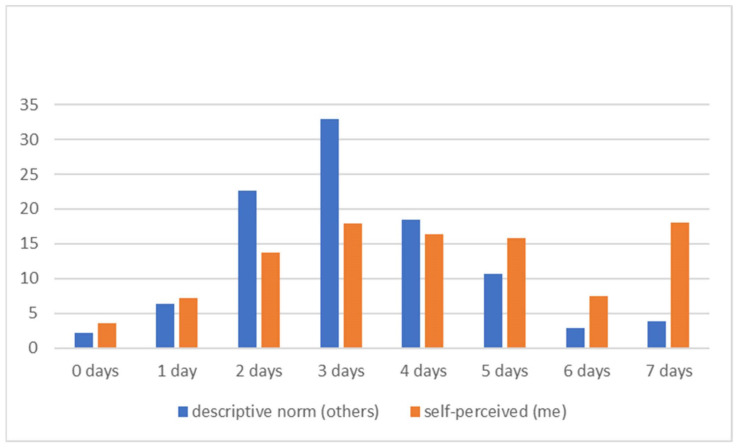
Misperceprion of MVPA among Czech adolescents.

**Figure 2 children-10-01134-f002:**
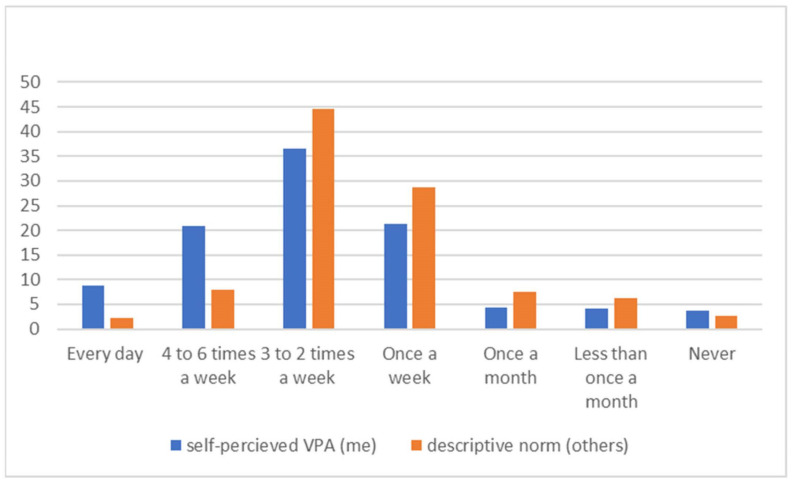
Misperception of VPA among Czech adolescents.

**Table 1 children-10-01134-t001:** Specific questions and responses used as physical activity measures.

Measured Domain	Questions	Responses
Moderate to vigorous physical activity over the past week (past week MVPA)	Over the past 7 days, on how many days were you physically active for a total of at least 60 min per day?	0 days; 1; 2; 3; 4; 5; 6; and 7 days
Descriptive social norm on past-week MVPA	Over the past 7 days, on how many days do you think most of your classmates were physically active for a total of at least 60 min per day?	0 days; 1; 2; 3; 4; 5; 6; and 7 days
Frequency of vigorous physical activity (VPA)	Outside school hours: How often do you usually exercise in your free time so much that you get out of breath or sweat?	Daily; 4–6 times a week; 2–3 times a week; Once a week; Once a month; Less than once a month; and Never
Descriptive social norm on VPA	Outside school hours: how often do you think most of your classmates usually exercise in their free time so much that you get out of breath or sweat?	None; About half an hour; About an hour; About 2–3 h; About 4–6 h; and 7 h or more

**Table 2 children-10-01134-t002:** Self-perceived MVPA among Czech adolescents according to gender and school class grade.

		0 Days		1 Day		2 Days		3 Days		4 Days		5 Days		6 Days		7 Days		Total	χ^2^
		*n*	%	*n*	%	*n*	%	*n*	%	*n*	%	*n*	%	*n*	%	*n*	%		(*p*)
Total		45	3.5%	93	7.2%	176	13.7%	231	17.9%	211	16.4%	204	15.8%	97	7.5%	232	18.0%	1289	
Boys		27	4.6%	30	5.1%	72	12.2%	97	16.4%	90	15.2%	105	17.7%	49	8.3%	122	20.6%	592	22.21
Girls		18	2.6%	63	9.0%	104	14.9%	134	19.2%	121	17.4%	99	14.2%	48	6.9%	110	15.8%	697	(0.002)
Grade	6th	15	4.3%	20	5.8%	35	10.1%	58	16.7%	63	18.2%	44	12.7%	29	8.4%	83	23.9%	347	
	7th	7	2.3%	25	8.0%	40	12.9%	46	14.8%	46	14.8%	59	19.0%	20	6.4%	68	21.9%	311	50.927
	8th	9	2.7%	20	6.0%	50	15.0%	62	18.6%	52	15.6%	58	17.4%	26	7.8%	56	16.8%	333	(<0.001)
	9th	14	4.7%	28	9.4%	51	17.1%	65	21.8%	50	16.8%	43	14.4%	22	7.4%	25	8.4%	298	

MVPA—moderate to vigorous physical activity.

**Table 3 children-10-01134-t003:** Perceived descriptive norms of MVPA among Czech adolescents according to gender and school class grade.

		0 Days		1 Day		2 Days		3 Days		4 Days		5 Days		6 Days		7 Days		χ^2^
		*n*	%	*n*	%	*n*	%	*n*	%	*n*	%	*n*	%	*n*	%	*n*	%	(*p*)
Total		27	2.1%	81	6.3%	293	22.7%	425	33.0%	239	18.5%	137	10.6%	37	2.9%	50	3.9%	
Boys		16	2.7%	34	5.7%	126	21.3%	175	29.6%	113	19.1%	77	13.0%	15	2.5%	36	6.1%	27.435
Girls		11	1.6%	47	6.7%	167	24.0%	250	35.9%	126	18.1%	60	8.6%	22	3.2%	14	2.0%	(<0.001)
Grade	6th	5	1.4%	25	7.2%	77	22.2%	108	31.1%	64	18.4%	38	11.0%	16	4.6%	14	4.0%	
	7th	8	2.6%	14	4.5%	65	20.9%	96	30.9%	64	20.6%	41	13.2%	7	2.3%	16	5.1%	31.954
	8th	7	2.1%	16	4.8%	84	25.2%	103	30.9%	62	18.6%	36	10.8%	10	3.0%	15	4.5%	(0.059)
	9th	7	2.3%	26	8.7%	67	22.5%	118	39.6%	49	16.4%	22	7.4%	4	1.3%	5	1.7%	

MVPA—moderate to vigorous physical activity.

**Table 4 children-10-01134-t004:** Underestimation of norms in moderate to vigorous physical activity (results of logistic regression).

			OR (95% C.I.)	*p*
Model 1	Gender	Boys	Ref.	
		Girls	1.469 (1.169–1.846)	0.001
Model 2	Gender	Boys	Ref.	
		Girls	1.468 (1.166–1.847)	0.001
	Grade	6th	Ref.	0.002
		7th	0.876 (0.640–1.200)	0.410
		8th	1.052 (0.770–1.437)	0.752
		9th	1.670 (1.193–2.339)	0.003
Model 3	Gender	Boys	Ref.	
		Girls	1.474 (1.171–1.855)	0.001
	Grade	6th	Ref.	0.003
		7th	0.875 (0.639–1.199)	0.407
		8th	1.043 (0.763–1.426)	0.790
		9th	1.647 (1.174–2.309)	0.004
	School		0.984 (0.954–1.015)	0.306
Model 4	Gender	Boys	Ref.	0.022
		Girls	1.336 (1.043–1.710)	
	Grade	6th	Ref.	0.191
		7th	0.831 (0.592–1.168)	0.287
		8th	0.932 (0.666–1.305)	0.682
		9th	1.239 (0.862–1.782)	0.247
	School		0.994 (0.961–1.027)	0.701
	MVPA		1.525 (1.425–1.632)	0.000

OR—odds ratio; MVPA—self-reported moderate to vigorous physical activity.

**Table 5 children-10-01134-t005:** Self-perceived VPA among Czech adolescents according to gender and school class grade.

		Every Day		4 to 6 Times a Week		3 to 2 Times a Week		Once a Week		Once a Month		Less than Once a Month		Never		Mean Rank	Test Value (*p*)
		*n*	%	*n*	%	*n*	%	*n*	%	*n*	%	*n*	%	*n*	%		
Total		115	8.9%	269	20.9%	470	36.5%	274	21.3%	57	4.4%	53	4.1%	48	3.7%		
Boys		64	10.8%	136	23.1%	218	36.9%	94	15.9%	27	4.6%	25	4.2%	26	4.4%	615.54	186,135 ^1^ (0.003)
Girls		51	7.3%	133	19.1%	252	36.2%	180	25.9%	30	4.3%	28	4.0%	22	3.2%	676.02	
Grade	6th	45	13.0%	74	21.4%	130	37.6%	61	17.6%	5	1.4%	18	5.2%	13	3.8%	693.98	27.929 ^2^ (<0.001)
	7th	30	9.7%	83	26.9%	110	35.6%	51	16.5%	13	4.2%	11	3.6%	11	3.6%	691.27	
	8th	27	8.1%	61	18.3%	121	36.3%	86	25.8%	16	4.8%	10	3.0%	12	3.6%	617.43	
	9th	13	4.4%	51	17.1%	109	36.6%	76	25.5%	23	7.7%	14	4.7%	12	4.0%	564.50	

^1^ Mann–Whitney U; ^2^ Kruskal–Wallis H.

**Table 6 children-10-01134-t006:** Perceived descriptive norms in VPA among Czech adolescents according to gender and school class grade.

		Every Day		4 to 6 Times a Week		3 to 2 Times a Week		Once a Week		Once a Month		Less than Once a Month		Never		Mean Rank	Test Value (*p*)
		*n*	%	*n*	%	*n*	%	*n*	%	*n*	%	*n*	%	*n*	%		
Total		28	2.2%	103	8.0%	575	44.7%	370	28.8%	97	7.5%	80	6.2%	33	2.6%		
Boys		19	3.2%	53	9.0%	271	45.9%	146	24.7%	47	8.0%	31	5.3%	23	3.9%	625.36	192,695 ^1^ (0.048)
Girls		9	1.3%	50	7.2%	304	43.7%	224	32.2%	50	7.2%	49	7.0%	10	1.4%	664.90	
Grade	6th	12	3.5%	42	12.1%	147	42.5%	98	28.3%	19	5.5%	21	6.1%	7	2.0%	691.31	22.478 ^2^ (<0.001)
	7th	9	2.9%	30	9.7%	150	48.5%	76	24.6%	18	5.8%	18	5.8%	8	2.6%	683.36	
	8th	5	1.5%	17	5.1%	154	46.2%	97	29.1%	31	9.3%	23	6.9%	6	1.8%	617.70	
	9th	2	0.7%	14	4.7%	124	41.6%	99	33.2%	29	9.7%	18	6.0%	12	4.0%	575.49	

^1^ Mann–Whitney U; ^2^ Kruskal–Wallis H.

**Table 7 children-10-01134-t007:** Underestimation of norms in vigorous physical activity (results of logistic regression).

			OR (95% C.I.)	*p*
Model 1	Gender	Boys	Ref.	
		Girls	1.274 (1.021–1.589)	0.032
Model 2	Gender	Boys	Ref.	
		Girls	1.273 (1.020–1.590)	0.033
	Grade	6th	Ref.	0.003
		7th	0.88 (0.643–1.204)	0.423
		8th	1.24 (0.915–1.68)	0.165
		9th	1.562 (1.143–2.135)	0.005
Model 3	Gender	Boys	Ref.	
		Girls	1.274 (1.020–1.592)	0.033
	Grade	6th	Ref.	0.003
		7th	0.88 (0.643–1.204)	0.423
		8th	1.238 (0.914–1.679)	0.168
		9th	1.558 (1.138–2.132)	0.006
	School	School	0.997 (0.968–1.027)	0.850
Model 4	Gender	Boys	Ref.	
		Girls	1.238 (0.981–1.563)	0.073
	Grade	6th	Ref.	0.056
		7th	0.885 (0.639–1.227)	0.464
		8th	1.161 (0.846–1.593)	0.357
		9th	1.387 (0.998–1.927)	0.051
	School		0.999 (0.968–1.03)	0.941
	VPA	Every day	Ref.	0.000
		4 to 6 times a week	0.493 (0.312–0.781)	0.003
		3 to 2 times a week	0.862 (0.568–1.309)	0.487
		Once a week	1.45 (0.927–2.269)	0.104
		Once a month	4.521 (2.148–9.518)	0.000
		Less than once a month	2.941 (1.465–5.905)	0.002
		Never	3.134 (1.513–6.489)	0.002

VPA—vigorous physical activity.

## Data Availability

The data presented in this study are available on request from the corresponding author. The data are not publicly available due to their containing information that could compromise the privacy of research participants.

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
