# Peer review of "How Czech Adolescents Perceive Their Physical Activity"

_children, 2023, doi:10.3390/children10071134_

Round 1
Reviewer 1 Report
Thanks for the opportunity to read this article.
The idea of the study is very interesting. After reading, I have some comments that I present below.
Line 28. Reference 1 is inappropriate, as it is a systematic review of academic performance. It should not be used to justify the benefits of physical activity.
The text has too many abbreviations. This makes reading difficult. I recommend removing abbreviations from the entire document. You can leave only the most used ones, such as “PA”.
Lines 91-117. The authors limited themselves to copying and pasting questions from a questionnaire. Questions should be presented and explained and not just pasted. For example, what were the response options?
There is no information about an opinion from an ethics board.
Line 124. What does it mean “Basic descriptive statistics”?
Lines 124-125. “Chi-squared tests and binomial regression models were” for what? Please explain what models were made.
Table 1. The table is confusing. What was compared about the “grades”? The number of days is a continuous variable. Why was Chi-square used?
Table 2. The table is confusing. What was compared about the “grades”? The number of days is a continuous variable. Why was Chi-square used?
Figure 1. The previous tables were stratified by gender. Why was this analysis not also stratified by sex?
Table 3. This table cannot be read. For example, “7 days” was used as a reference, but this variable is continuous and could have been included in the analysis in its continuous form.
I have many reservations about all the analyzes made in this article. Chi-Square is not the most suitable test for the data used in most cases.
The first paragraph of the discussion should serve to recall the purpose of the study and highlight the main results. Authors use this paragraph to talk about other authors.
Author Response
1. Thanks for the opportunity to read this article.
The idea of the study is very interesting. After reading, I have some comments that I present below.
Thank you.
2. Line 28. Reference 1 is inappropriate, as it is a systematic review of academic performance. It should not be used to justify the benefits of physical activity.
Thank you for suggestion. The reference was replaced with more appropriate one. (van Sluijs, E.M.F.; Ekelund, U.; Crochemore-Silva, I.; Guthold, R.; Ha, A.; Lubans, D.; Oyeyemi, A.L.; Ding, D.; Katzmarzyk, P.T. Physical Activity Behaviours in Adolescence: Current Evidence and Opportunities for Intervention. The Lancet 2021, 398, 429–442, doi:10.1016/S0140-6736(21)01259-9.)
3. The text has too many abbreviations. This makes reading difficult. I recommend removing abbreviations from the entire document. You can leave only the most used ones, such as “PA”.
Thank you for suggestion to improve the readability of our manuscript. We removed the following abbreviations FTNC, TPB, TRA SNA. We decided to keep PA, MVPA and VPA due the frequency of use.
4. Lines 91-117. The authors limited themselves to copying and pasting questions from a questionnaire. Questions should be presented and explained and not just pasted. For example, what were the response options?
Although response options were presented in the manuscript, we appreciate the suggestion to make this part more readable. Information about reliability and validity was added and the description of item is presented in tabular form.
|
Table 1 Specific questions and responses used as physical activity measures |
||
|
Measured domains |
Questions |
Responses |
|
Moderate to vigorous physical activity over the past week (past week MVPA) |
Over the past 7 days, on how many days were you physically active for a total of at least 60 min per day? |
0 days; 1; 2; 3; 4; 5; 6; 7 days |
|
Descriptive social norm on past week MVPA |
Over the past 7 days, on how many days do you think most of your classmates were physically active for a total of at least 60 minutes per day? |
0 days; 1; 2; 3; 4; 5; 6; 7 days |
|
Frequency of vigorous physical activity (VPA) |
Outside school hours: How often do you usually exercise in your free time so much that you get out of breath or sweat? |
Daily; 4–6 times a week; 2–3 times a week; Once a week; Once a month; Less than once a month; Never |
|
Descriptive social norm on VPA |
Outside school hours: how often do you think most of your classmates usually exercise in their free time so much that you get out of breath or sweat?” |
None; About. half an hour; About. an hour; About. 2–3 h; About 4–6 h; 7 hours or more |
5. There is no information about an opinion from an ethics board.
Information from ethics board was presented at the end of the article. As more reviewers struggled to find the information we moved it to 2.1. Procedure Sample and Participant Selection.
6. Line 124. What does it mean “Basic descriptive statistics”?
2.4. Data Processing was thoroughly rewritten to be more informative and readable. We do not use the term now.
7. Lines 124-125. “Chi-squared tests and binomial regression models were” for what? Please explain what models were made.
Thank you for valuable suggestion. 2.4. Data Processing was thoroughly rewritten to be more informative and readable.
"2.4. Data Processing
To describe the gender and school grade differences, Independent sample T test and ANOVA was performed for MVPA and Mann–Whitney U test Kruskal–Wallis H test for VPA. To analyze the effect of gender, school grade, school and participants own PA on underestimation of PA a binomial regression models were used. Sample was dichotomized into two groups: underestimated PA (1) vs. rest of the sample (0). In four steps we entered sex, grade, school and own PA. Statistical data processing was performed using the IBM SPSS Statistics software, version 23 (IBM, Armonk, NY, USA)."
8. Table 1. The table is confusing. What was compared about the “grades”? The number of days is a continuous variable. Why was Chi-square used?
Thank you for honest and constructive suggestions. We re-analysed the data and do not use chi-square anymore. More details please see point 7 above. The tables and text were reformulated according new results.
9. Table 2. The table is confusing. What was compared about the “grades”? The number of days is a continuous variable. Why was Chi-square used?
Thank you for honest and constructive suggestions. We re-analysed the data and do not use chi-square anymore. More details please see point 7 above. The tables and text were reformulated according new results.
10. Figure 1. The previous tables were stratified by gender. Why was this analysis not also stratified by sex?
The intention with using the graphs (Figure 1 and 2) was to visualise the misperception of PA in general. We found that pattern of misperception similar to that on general level appears also on gender and school grade level. Thus we did not prepared a graphs for gender and school grade.
We slightly reformulated the description of results to make the results more readable.
"When comparing self-perceived MVPA and perceived descriptive norm in MVPA (i.e. adolescents own behaviour vs. behaviour of their peers) we found clear misperception. (Graph 1) Adolescents perceive the level of MVPA of their peers as much lower. While majority of adolescent (57.7%) have reported 60 minutes of MVPA at least 4 times a week, only 35.9% believe that their peers have 60 minutes of MVPA at least 4 times a week. The pattern of missperception is present both for boys and girls and across all grades (Table 1 and Table 2)."
11. Table 3. This table cannot be read. For example, “7 days” was used as a reference, but this variable is continuous and could have been included in the analysis in its continuous form.
Thank you for valuable suggestion. Data was reanalysed as suggested, table was updated. Results were not affected in significant way.
12. I have many reservations about all the analyzes made in this article. Chi-Square is not the most suitable test for the data used in most cases.
Thank you for honest and constructive suggestions. We re-analysed the data and do not use chi-square anymore. More details in point 7 above.
13. The first paragraph of the discussion should serve to recall the purpose of the study and highlight the main results. Authors use this paragraph to talk about other authors.
Thank you for your valuable suggestion, the following paragraph was added at the beginning of discussion.
"Our research highlights a disparity between individuals' own perceptions of their physical activity levels and their perceptions of their peers' physical activity levels. Adolescents tend to underestimate the prevalence of adequate PA among their peers, resulting in perceived descriptive norms that indicate lower levels of physical activity compared to their own. Girls and older students demonstrate a notable tendency to underestimate the prevalence of physical activity among their peers. Interestingly, the school environment does not appear to have an impact on the perception of PA. When individuals' own self-perceived levels of physical activity (PA) are taken into account, it becomes evident that they explain a significant portion of the variability in misperception related to gender and school grade. Considering self-perceived PA tends to account for nearly all of the differences observed among genders and school grades in terms of misperception."

Reviewer 2 Report
Abstract: By what method were MVPA and VPA found? It should be written in the abstract.
What is meant by social norms? Line 21 and Line 259.
Keywords: "Children" is written, but there is no case about children in the study (Adolescent). It should be removed.
Line 73-79: The importance of the manuscript should be mentioned.
Line 88-90: Information about HSBC questions should be given. Validity and reliability.
As for the Discussion, this section needs to be expanded. For example, the authors did not discuss the gender difference in their results. Appropriate comparisons with the literature should also be made.
Minor editing of English language.
Author Response
1. Abstract: By what method were MVPA and VPA found? It should be written in the abstract.Abstract was updated. The following sentence was added: "PA was measured using self-reported items used in HBSC study."
2. What is meant by social norms? Line 21 and Line 259.
Thank you for suggestion. In order to better define/describe social norms for readers, the following paragraph was added as second paragraph of introduction.
"The conduct of individuals regarding their health is significantly impacted by social norms. Social norms refer to the shared standards, expectations, and rules of behavior that are considered acceptable and appropriate within a particular social group or society. These norms guide and regulate individual and group behavior, defining what is considered normal, acceptable, and desirable in terms of actions, attitudes, values, and beliefs. Thus perceiving of what is “normal” influences individuals behaviour. Such influence ranges from reinforcing favourable behaviours that can safeguard and improve one's well-being, to promoting unfavourable actions that elevate the possibility of detrimental health outcomes.[4]"
3. Keywords: "Children" is written, but there is no case about children in the study (Adolescent). It should be removed.
Thank you for suggestion, the keyword was removed.
4. Line 73-79: The importance of the manuscript should be mentioned.
Thank you for suggestion. To stress the importance of manuscript ee restructured the paragraph in Introduction and in addition expanded "Strenghts and limitation" in Discussion.
"Gaining insights into how adolescents perceive their physical activity is essential for the development of effective interventions and strategies aimed at promoting a more active lifestyle. Understanding the disparity between individuals' self-perceived levels of physical activity and their perception of social norms is particularly important when applying the social norms approach. Previous research has devoted relatively limited attention to this specific area, underscoring the significance of our study as a valuable addition to the existing literature. By addressing this research gap, our study provides novel insights and contributes to a more comprehensive understanding of the subject matter. "
"Nevertheless, this study emphasizes the importance of examining adolescents' physical activities within the context of their perceptions of social norms related to physical activity. This specific area has received relatively less attention in previous research, making this study a valuable contribution to the existing literature. To the best of our knowledge, no study investigating this specific topic has been conducted in the Czech Republic or Central Europe. Thus, our study fills an important research gap by being the first of its kind in this region, providing valuable insights into the perceptions of adolescents' physical activities and their relationship with social conventions. By exploring this unexplored territory, our research expands the existing knowledge base and contributes to a more comprehensive understanding of the factors influencing adolescents' physical activity behaviors in the Czech Republic and Central Europe."
5. Line 88-90: Information about HSBC questions should be given. Validity and reliability.
Thank you for suggestion. Information about reliability and validity was added. For the purposes of this study, we used questions on MVPA and VPA from HBSC study.[21] According to current systematic review[22] PA measures of the HBSC questionnaire are moderately reliable (ICC 0.41–0.6). When assessing validity, the HBSC PA measures show fair to moderate performance (r 0.21-0.6).
6. As for the Discussion, this section needs to be expanded. For example, the authors did not discuss the gender difference in their results. Appropriate comparisons with the literature should also be made.
Thank you for the valuable suggestion. The discussion was significantly updated. With regards gender differences whole new paragraph was added.
"In line of numerous studies before,[29,30] our findings confirmed that girls report lower levels of PA. In addition, according our findings girls tend to significantly more underestimate the prevalence of VPA among their peers. In their comprehensive review of existing systematic reviews, Duffey et al. [31] examine the barriers and facilitators of PA specifically in adolescent girls. Their findings indicate that the most commonly reported barriers to PA among adolescent girls were the absence of support from peers, family, and teachers, closely followed by time constraints. On the other hand, the most frequently identified facilitators of PA were weight loss/management goals and the support provided by peers, family, and teachers. It is worth noting that the majority of factors influencing PA engagement among girls primarily originated from the individual and interpersonal levels. Our results confirm their findings and indicates that social norms are important component in facilitating PA among adolescent girls."

Reviewer 3 Report
Since gender and class grade are among their objectives, they should provide more information about what the theory says about these variables in relation to the subject of study.
In methodology, I think the inclusion and exclusion criteria should be more specified.
Has the research received approval from an ethics committee?
The results seem interesting to me, congratulations.
I think that the discussion should work more on what was found on the misperception (only 35.9% believe that their peers have 60 minutes of MVPA at least 4 times a 146 week).
In the discussion, the contribution they provide with their research should be highlighted and worked on, and the benefits it brings to practice.
Congratulations, you did a good job, just need a few tweaks.
Author Response
1 Since gender and class grade are among their objectives, they should provide more information about what the theory says about these variables in relation to the subject of study.
Thank you for suggestion, the following text was added into Introduction.
"Gender differences in physical activity among adolescent boys and girls are multifaceted and intertwined with societal norms, gender roles, and self-presentation.[18] As they transition from childhood to adolescence, peer influence increases and plays a crucial role in their engagement in physical activity.[19] This influence can comprise social support, the presence of peers during physical activity, peer norms, friendship quality, changes to friendship groups, preferences for certain activities among peers, and affiliation to peer groups.[20] Additionally, potential experiences of peer victimization may further limit girls’ participation in PA.[21] Despite being central factors affecting girls' physical activity involvement, the influence of peers has often been overlooked by many interventions aimed at promoting PA.[18]"
2 In methodology, I think the inclusion and exclusion criteria should be more specified.
2.1. Procedure Sample and Participant Selection part was thoroughly rewritten. We hope that it is ore informative in readable now.
"Our cross-sectional study involved 1,289 adolescents from the Czech Republic. The participants were aged between 11 and 15 years, with a mean age of 13.49 years and a standard deviation of 1.16 years. Out of the total participants, 46% were boys. The study included 12 randomly selected schools, which comprised students from Grades 6 to 9. Schools specializing in sports and schools for students with special educational needs were not included in the study. Within each school we randomly selected one class per grade. It was assumed that the classes within each school were approximately equal in size. This sampling approach aimed to create a self-weighted sample, wherein each student had an approximately equal chance of being selected for participation.
Participation in the study was voluntary, meaning that the adolescents could choose whether or not to participate. No incentives were provided to encourage participation, and the data collection process did not involve any form of compensation or rewards. Additionally, the study reported that less than five percent of the adolescents opted out of participating, indicating a relatively high rate of participation among the targeted population.
The Ethics Committee of the Faculty of Physical Culture at Palacký University Olomouc approved the study On May 9th, 2017, with reference number 38/17. The committee adheres to the ethical standards outlined in the World Medical Association Declaration of Helsinki and its subsequent amendments."
3 Has the research received approval from an ethics committee?
Information from ethics board was presented at the end of the article. As more reviewers struggled to find the information we moved it to 2.1. Procedure Sample and Participant Selection.
4 The results seem interesting to me, congratulations.
Thank you
5 I think that the discussion should work more on what was found on the misperception (only 35.9% believe that their peers have 60 minutes of MVPA at least 4 times a 146 week).
Part of the the discussion was restructured and some new text was added in line of this valuable suggestion.
"Our study highlights a notable disparity between individuals' self-perceived levels of moderate-to-vigorous physical activity (MVPA) and vigorous physical activity (VPA) and their perceptions of their peers' physical activity norms. Despite the majority of adolescents (57.7%) reporting their engagement in at least four 60-minute sessions of MVPA per week, a significantly lower percentage (35.9%) perceive their peers as achieving the same activity level. Importantly, this pattern of misperception extends to VPA as well, affecting both boys and girls across all grades. These findings have significant implications for the utilization of social norms in intervention strategies aimed at promoting sufficient physical activity among adolescents. Perkins[23] has identified two pre-conditions that must be met before implementing social norms approach properly. The first condition necessitates a discrepancy between perceived and actual behaviour, which indicates overestimation of problem conduct. The second condition demands that at least half of the population behaves responsibly; given that individuals strive to conform with societal norms, employing a social norms message campaign may inadvertently encourage harmful conduct if most people engage in it. Hence, if more than 50% violate desirable behaviours targeted by the intervention program, recognizing social norms approach limitations would prove critical in decision-making regarding its implementation potentiality.[17,24]"
6. In the discussion, the contribution they provide with their research should be highlighted and worked on, and the benefits it brings to practice.
Thank you for suggestion. The following paragraph was added into the discussion part:
"Nevertheless, this study emphasizes the importance of examining adolescents' physical activities within the context of their perceptions of social norms related to physical activity. This specific area has received relatively less attention in previous research, making this study a valuable contribution to the existing literature. To the best of our knowledge, no study investigating this specific topic has been conducted in the Czech Republic or Central Europe. Thus, our study fills an important research gap by being the first of its kind in this region, providing valuable insights into the perceptions of adolescents' physical activities and their relationship with social conventions. By exploring this unexplored territory, our research expands the existing knowledge base and contributes to a more comprehensive understanding of the factors influencing adolescents' physical activity behaviors in the Czech Republic and Central Europe."
7. Congratulations, you did a good job, just need a few tweaks.
Thank you.

Round 2
Reviewer 1 Report
I have seen and acknowledged that the authors have made a great effort to improve the article. However, I have the same reservations about the data analysis. For this reason, I maintain my position on rejecting the article.